# A Comparative Study of Different Machine Learning Algorithms in Predicting the Content of Ilmenite in Titanium Placer

Yingli LV [1], Qui-Thao Le [2,3], Hoang-Bac Bui [4,5], Xuan-Nam Bui [2,3], Hoang Nguyen [6,*], Trung Nguyen-Thoi [7,8,*], Jie Dou [9,*] and Xuan Song [10]

1   Department of Electrical Engineering, Jiyuan Vocational and Technical College, Jiyuan 459000, China; lvyinglisunny@126.com
2   Department of Surface Mining, Mining Faculty, Hanoi University of Mining and Geology, 18 Vien st., Duc Thang ward, Bac Tu Liem dist., Hanoi 100000, Vietnam; lequithao@humg.edu.vn (Q.-T.L.); buixuannam@humg.edu.vn (X.-N.B.)
3   Center for Mining, Electro-Mechanical research, Hanoi University of Mining and Geology, 18 Vien st., Duc Thang ward, Bac Tu Liem dist., Hanoi 100000, Vietnam
4   Faculty of Geosciences and Geoengineering, Hanoi University of Mining and Geology, 18 Vien st., Duc Thang ward, Bac Tu Liem dist., Hanoi 100000, Vietnam; buihoangbac@humg.edu.vn
5   Center for Excellence in Analysis and Experiment, Hanoi University of Mining and Geology, 18 Vien st., Duc Thang ward, Bac Tu Liem dist., Hanoi 100000, Vietnam
6   Institute of Research and Development, Duy Tan University, Da Nang 550000, Vietnam
7   Division of Computational Mathematics and Engineering, Institute for Computational Science, Ton Duc Thang University, Ho Chi Minh City 700000, Vietnam
8   Faculty of Civil Engineering, Ton Duc Thang University, Ho Chi Minh City 700000, Vietnam
9   Civil and Environmental Engineering, Nagaoka University of Technology, 1603-1, Kami-Tomioka, Nagaoka, Niigata 940-2188, Japan
10   Center for Spatial Information Science, the University of Tokyo, 5-1-5, Kashiwa 277-8568, Japan; songxuan@csis.u-tokyo.ac.jp
*   Correspondence: nguyenhoang23@duytan.edu.vn (H.N.); nguyenthoitrung@tdtu.edu.vn (T.N.-T.); douj888@gmail.com (J.D.)

**Abstract:** In this study, the ilmenite content in beach placer sand was estimated using seven soft computing techniques, namely random forest (RF), artificial neural network (ANN), *k*-nearest neighbors (kNN), cubist, support vector machine (SVM), stochastic gradient boosting (SGB), and classification and regression tree (CART). The 405 beach placer borehole samples were collected from Southern Suoi Nhum deposit, Binh Thuan province, Vietnam, to test the feasibility of these soft computing techniques in estimating ilmenite content. Heavy mineral analysis indicated that valuable minerals in the placer sand are zircon, ilmenite, leucoxene, rutile, anatase, and monazite. In this study, five materials, namely rutile, anatase, leucoxene, zircon, and monazite, were used as the input variables to estimate ilmenite content based on the above mentioned soft computing models. Of the whole dataset, 325 samples were used to build the regarded soft computing models; 80 remaining samples were used for the models' verification. Root-mean-squared error (RMSE), determination coefficient ($R^2$), a simple ranking method, and residuals analysis technique were used as the statistical criteria for assessing the model performances. The numerical experiments revealed that soft computing techniques are capable of estimating the content of ilmenite with high accuracy. The residuals analysis also indicated that the SGB model was the most suitable for determining the ilmenite content in the context of this research.

**Keywords:** titanium placer; beach placer; ilmenite content; artificial intelligence; applied soft computing

## 1. Introduction

In the context of the intense development of the world, the demand for machinery, paint, paper, and plastic is virtually unending with titanium minerals labelled as one of the primary materials [1–3]. This commodity can be exploited from the placer and hard rocks deposits [4]. In particular, placer deposits are abundant sources of titanium for coastal countries and Vietnam is one of those countries [5]. They are considered to be economically valuable minerals because of their ease of exploitation and flexibility [6]. In titanium placer, the major mineral components are ilmenite, rutile, anatase, leucoxene, zircon, and monazite [7]. Of those minerals, ilmenite and rutile are normally considered as the main types containing titanium because of their high proportion in the total heavy minerals [8,9]. Although the concentration of $TiO_2$ is higher in natural rutile than ilmenite, the distribution of rutile is normally inadequate for processing in titanium placer. Furthermore, sulfate and chloride processes can be applied as the chemical methods to obtain higher $TiO_2$ content [10,11]. In other words, $TiO_2$ can be enriched from ilmenite by chemical methods [12–15]. Therefore, ilmenite can be considered as the main mineral used to extract $TiO_2$ in titanium placer.

In recent years, mineral exploration based on data-driven or artificial intelligence (AI) techniques is considered as a cost-effective alternative to traditional methods. The spatial and geochemical datasets have been analyzed by traditional analytical or AI approaches with high reliability [16–20]. Mineralization, as well as minerals, can be forecasted using these techniques to highlight mineral potential based on similar data attributes [21]. For example, the ant colony algorithm was performed to recognize the geochemical anomalies in the interpolated concentration of Au, Cu, Ag, Zn, and Pb by Chen, An [22]. The artificial neural network (ANN), *k*-nearest neighbors (kNN) models were also developed to investigate the components of mineral of coal and maceral groups by Mlynarczuk, Skiba [23]. To evaluate the relationship between mineral potentials as well as establish mineral prospectivity maps, Maepa, Smith [24] used ANN, fuzzy logic, and logistics regression as powerful tools for the mapping of potential gold deposits. Zuo, Xiong [25] also deployed several machine learning (ML) techniques to identify geochemical anomalies of Fe polymetallic deposits. They concluded that ML techniques are robust tools for discovering multivariate geochemistry anomalies. In another study, Johnson et al. [18] used ANN with 96 data points to evaluate the distribution of geochemical of gold deposits at Canning Basin (Western Australia). Their results showed that ANN was a feasible technique for assessing geochemical property distribution with a determination coefficient ($R^2$) of 0.8. To establish another approach to geochemical mapping, Zuo et al. [26] applied deep learning and indicated that this method could deal with nonlinear and complex problems. In addition, various ML techniques have been used to detect the content as well as the potential of minerals and geochemical anomalies as the following works [18,25,27–34].

Review of the published works shows that the mapping of mineral distribution can be achieved by using data-driven techniques. Although a number of studies have been carried out to model several commodities, such as Fe, Au, Zn, Cu, there is still a scarcity of research on titanium in general and ilmenite in particular. Motivated from the significance of this type of mineral on the local economy [35], this study aimed at evaluating the feasibility of predicting it using different AI techniques. From extensive review of previous works, seven different AI methods presenting four groups were selected: Decision tree algorithms group (random forest (RF) and classification and regression tree (CART), boosting algorithms group (stochastic gradient boosting (SGB)), neural networks group (ANN), and nonlinear algorithms groups (support vector machine (SVM), cubist, and kNN). Based on the obtained results, the best method will be introduced as the state-of-the-art technique for predicting ilmenite content.

## 2. Background of Artificial Intelligence Techniques Used

### 2.1. Random Forest

As described by Breiman [36] first, RF is a decision tree algorithm in statistical communication. It is known as a useful tool for classification and regression issues. Inspired from an election, each decision tree acts as a voter. The set of all votes for the final decision is used to improve the predictions' accuracy [37–39]. The background of the RF algorithm can be described according to the following pseudo-code (Figure 1).

---

**Algorithm 1: Pseudo code for the random forest algorithm**

To generate $c$ bootstrap samples:
**for** $i = 1$ to $c$ **do**
    Randomly sample the training data $D$ with replacement to produce $D_i$
    Create a root node, $N_i$ containing $D_i$
    Call BuildTree($N_i$)
**end for**

**BuildTree(N):**
**if** $N$ contains instances of only one class **then**
    **return**
**else**
    Randomly select x% of the possible splitting features in $N$
    Select the feature $F$ with the highest information gain to split on
    Create f child nodes of $N$, $N_1,..., N_f$, where $F$ has $f$ possible values $(F_1, … , F_f)$
    **for** $i = 1$ to $f$ **do**
        Set the contents of $N_i$ to $D_i$, where $D_i$ is all instances in $N$ that match
        $F_i$
        Call BuildTree($N_i$)
    **end for**
**end if**

---

**Figure 1.** Random forest's (RF) pseudo-code [40].

### 2.2. Stochastic Gradient Boosting

SGB is one of the ensemble techniques proposed by Friedman [41]. Based on decision tree algorithm [42,43], SGB was improved by using boosting learning and editing error of decision trees. Like RF technique, SGB can solve all classification, as well as regression issues. The theory of SGB algorithm is shown in Figure 2.

### 2.3. CART

In data mining, CART was introduced as an effective nonparametric algorithm for forecasting issues, including regression and classification. Additionally, it is also known as a robust decision tree algorithm for forecasting problems [44]. Inspired by the development of trees in nature, CART's operational principles are developed based on mapping data [45]. Variables in data are represented by internal nodes (i.e., rutile, anatase, leucoxene, zircon, and monazite). The leaf nodes represent the outcomes (i.e., ilmenite content).

Unlike other techniques, CART does not require data normalization and can work well with outliers [46]. Furthermore, the CART algorithm can clearly explain situations as a "white box algorithm" [47]. Additionally, statistical tests can be applied to model verification to increase their reliability. More details of the CART algorithm can be found in [48–51].

---

**Algorithm 2: Pseudo code for the Stochastic gradient boosting algorithm**

1      $F_0(x) = \arg\min_\gamma \sum_{i-1}^N \psi(y_i, \gamma)$

2      **For** m=1 to **M do**:

3      $\widetilde{y}_{im} = -\left[\dfrac{\partial \psi(y_i, F(x_i))}{\partial F(x_i)}\right]_{F(x) = F_{m-1}(x)}, i = 1, N$

4      $\{R_{lm}\}_1^L = L - \text{terminal node tree}\left(\{\widetilde{y}_{im}, x_i\}_1^N\right)$

5      $\gamma_{lm} = \arg\min_\gamma \sum_{x_i \in R_{lm}} \psi(y_i, F_{m-1}(x_i) + \gamma)$

6      $F_m(x) = F_{m-1}(x) + \nu \cdot \gamma_{lm} 1(x \in R_{lm})$

7      **endFor**

---

**Figure 2.** Pseudo-code of the stochastic gradient boosting (SGB) technique. Reproduced with permission from [41], Copyright Elsevier, 2002.

## 2.4. SVM

SVM is well known as a benchmark AI technique in the statistical community. It can be applied for forecasting/predicting any regression/classification issues [52]. The SVM theory is based on the minimization of structural risk [53,54].

For regression problems, the functions of the kernel are often to be used to predict resulting outcome, such as radial basis function (RBF), two neural networks, polynomial, sigmoid and linear, exponential radial basis function (ERBF) [55,56]. In recent years, SVM has been applied in many fields as well as publications, therefore, the details of the SVM are not presented in this study but can be found in [57–63].

## 2.5. Cubist

As well-developed rule-based model, Cubist algorithm (CA) was proposed by Quinlan [64] and widely introduced by Rulequest [65]. It works on the idea of the nearest neighbors in the set of training data with additional corrections [66]. Like the M5′ Rules model, the CA can generate the rules for forecasting classification and regression issues [67]. On the other hand, CA is classified as a decision tree technique. Although it also creates an initial tree as the first step, however, unlike the M5 tree model, the rule is generated by the pruning the tree (collapses the paths). For regression issues (e.g., prediction of ilmenite content), the dataset is defined by the rules, and the CA model can fit for each rule. To avoid overfitting in CA, the rules can be combined or pruned. Subsequently, split-caused pruned are smoothed to compensate for the sharp discontinuities. Briefly, the CA can be described in four steps, as illustrated in Figure 3.

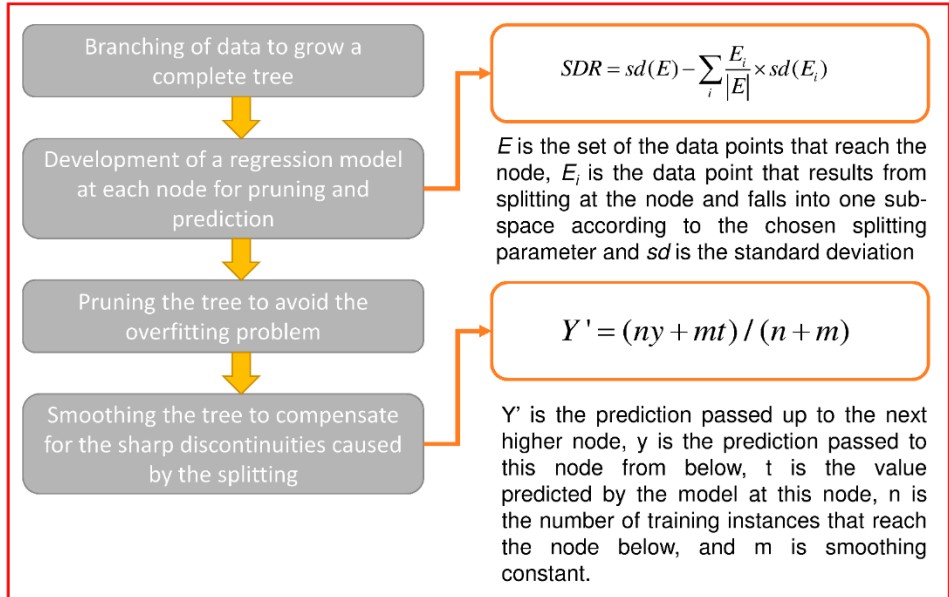

**Figure 3.** Cubist model and its operation for estimating ilmenite component.

## 2.6. The k-Nearest Neighbors

In machine learning, kNN is classified as a lazy algorithm [68,69]. It stores all observations that it reads then predicts other observations based on distance functions. The goal of kNN is to compute a numerical target averaged from the *k* nearest neighbors [70]. In addition, the inverse distance weighted average is also used for computing such distance [71]. For regression problems, kNN uses three distance functions to compute the distance between the neighbors as follow:

$$\text{Euclidean function}: \sqrt{\sum_{i=1}^{f}(x_i - y_i)^2} \tag{1}$$

$$\text{Manhattan function}: \sum_{i=1}^{f}|x_i - y_i| \tag{2}$$

$$\text{Minkowski function}: \left(\sum_{i=1}^{f}\left(|x_i - y_i|\right)^q\right)^{1/q} \tag{3}$$

where $x_i$ and $y_i$ are the *i*th-dimensions of the *x* and *y* points; *q* is the order between two points *x* and *y*.

## 2.7. ANN

As a model of information processing, ANN was introduced and simulated based on the idea of the human brain [72]. It is capable of processing information quickly and accurately based on the connections of neurons. In fact, ANN is also considered to be smarter than humans as they are capable of energetic calculations and self-evolution [73]. ANN can learn problems quickly and remember them. Then, based on the experience acquired, it can predict new observations [74].

There are many types of ANN, such as multilayer perceptron neural network (MLP), recurrent neural network (RNN), and convolutional neural network (CNN) to name a few. However, MLP is still a popular technique due to its simplicity and efficiency [75,76]. Therefore, this study used an MLP-type ANN model. Accordingly, its structure consists of three parts: Input layer (i.e., rutile, anatase, leucoxene, zircon, and monazite), hidden layers, and output layer (i.e., ilmenite content).

The operation method of ANN for estimating the ilmenite content is as follows:

Step 1: The input neurons receive signals from the external environment (the weight percent of each heavy mineral: Rutile, anatase, leucoxene, zircon, and monazite).

Step 2: Calculate weights and biases.

Step 3: Send information that has been preprocessed to the first hidden layer. Transfer functions can be enabled to transmit information between layers.

Step 4: Perform learning and calculation in the first hidden layer.

Step 5: Recalculate weights and biases after learning in the first hidden layer.

Step 6: Send the results to the second hidden layer,

Step 7: Perform the same actions as done in the first hidden layer.

Step 8: Send the calculation results, weights, and biases in the second hidden layer to the output layer.

Step 9: Repeat the same calculations for the next hidden layer.

Step 10: Estimate the ilmenite content and produce the final result.

## 3. Data Collection

The study area is the Southern Suoi Nhum titanium placer deposit, Binh Thuan province (Vietnam), as shown in Figure 4. The previous geological surveys indicated that the study area and surroundings have loess sediments of the Pleistocene to Holocene age. These Quaternary sediments are distributed into sandy strips running parallel to the coastline. Exploration results showed that the mineral components of the placer sand consist mainly of ilmenite, rutile, anatase, leucoxene, zircon, and monazite (Figure 5). Among these heavy minerals, ilmenite is a common mineral and accounts for a significant proportion. These minerals exist in the red marine sediment of the Pleistocene age (Phan Thiet formation) and a gray marine-eolian residue of the Holocene age. Deo Ca Complex can be found in the surrounding areas. The complex consist of whitish-gray grano-syenite, biotite granite, and biotite-hornblende granite [35].

For data collection, placer sand samples were dried and mixed thoroughly in the laboratory. The samples were then sieved using the 1.18 mm (American Society for Testing and Materials) ASTM sieve. The coning and quartering method was applied continuously to reduce the sample to 20–30 g. The ultrafine clays in the samples were removed using distilled water. After drying, the total heavy minerals (THM) were isolated by using bromoform heavy liquid. The magnetic/nonmagnetic heavy materials in THM were separated using hand magnets. Then, each type of heavy mineral (ilmenite, rutile, anatase, leucoxene, zircon, and monazite) was determined by manual grain counting using an optical microscope. Finally, the weight percentage of individual heavy minerals was calculated by multiplying their percentage with the respective specific gravity values. In this study, 405 samples were collected from different positions of the mine. Scanning electron microscope ((SEM) Quanta 450, FEI company) with energy-dispersive X-ray spectroscopy (EDS) was used to double-check the identified heavy minerals through their morphology and composition. In order to prepare the samples for SEM-EDS analysis, the heavy minerals were put onto the surface of carbon conductive tape that attached to the specimen stub. The samples were then coated by carbon to enhance the quality of SEM images.

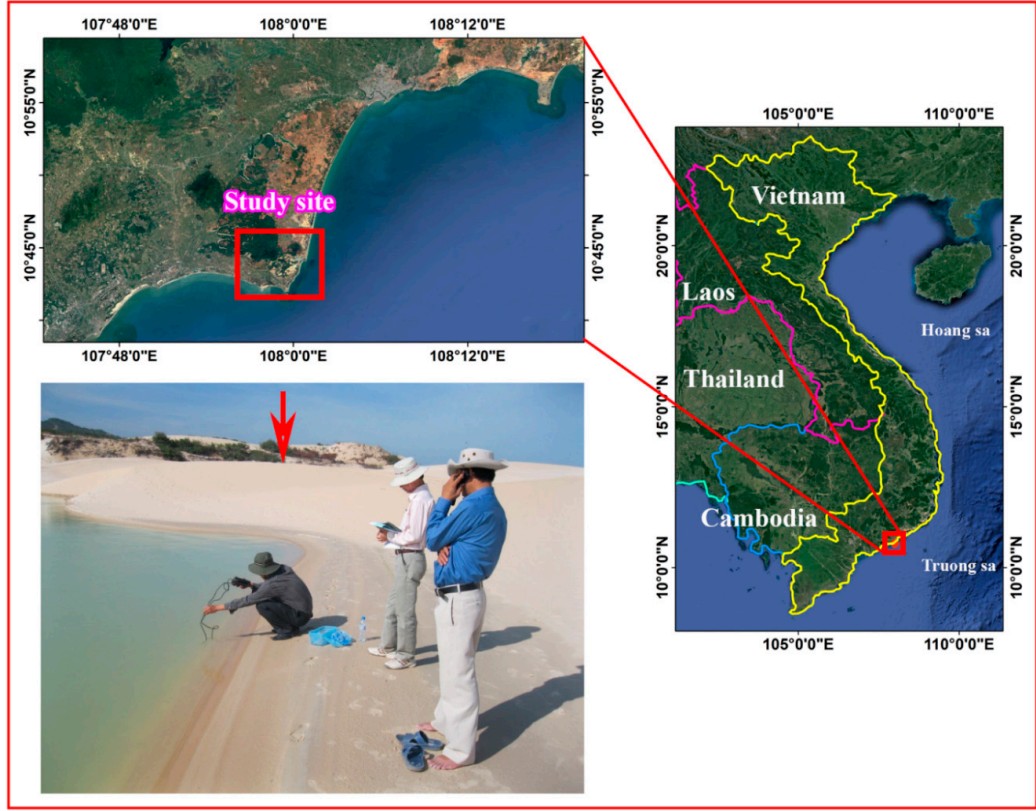

**Figure 4.** A view of the Southern Suoi Nhum titanium placer mine (Vietnam).

In this study area, ilmenite is a common mineral and accounts for a significant proportion. For AI calculation, the weight percent of each heavy mineral (i.e., rutile, anatase, leucoxene, zircon, and monazite) were used as input data for predicting ilmenite content (not titanium content in each heavy mineral). The statistical properties of the heavy minerals are listed in Table 1.

**Table 1.** Statistical attributes of the weight percent of each heavy mineral.

| Classification | Rutile | Anatase | Leucoxene | Zircon | Monazite | Ilmenite |
|:---:|:---:|:---:|:---:|:---:|:---:|:---:|
| Min.: | 0.000 | 0.000 | 0.000 | 0.013 | 0.001 | 0.188 |
| 1st Qu.: | 0.001 | 0.0001 | 0.003 | 0.030 | 0.001 | 0.306 |
| Median: | 0.001 | 0.0001 | 0.005 | 0.052 | 0.001 | 0.424 |
| Mean: | 0.001343 | 0.0001573 | 0.008 | 0.064 | 0.003 | 0.467 |
| 3rd Qu.: | 0.002 | 0.0002 | 0.012 | 0.080 | 0.002 | 0.538 |
| Max.: | 0.009 | 0.0007 | 0.032 | 0.306 | 0.058 | 2.246 |

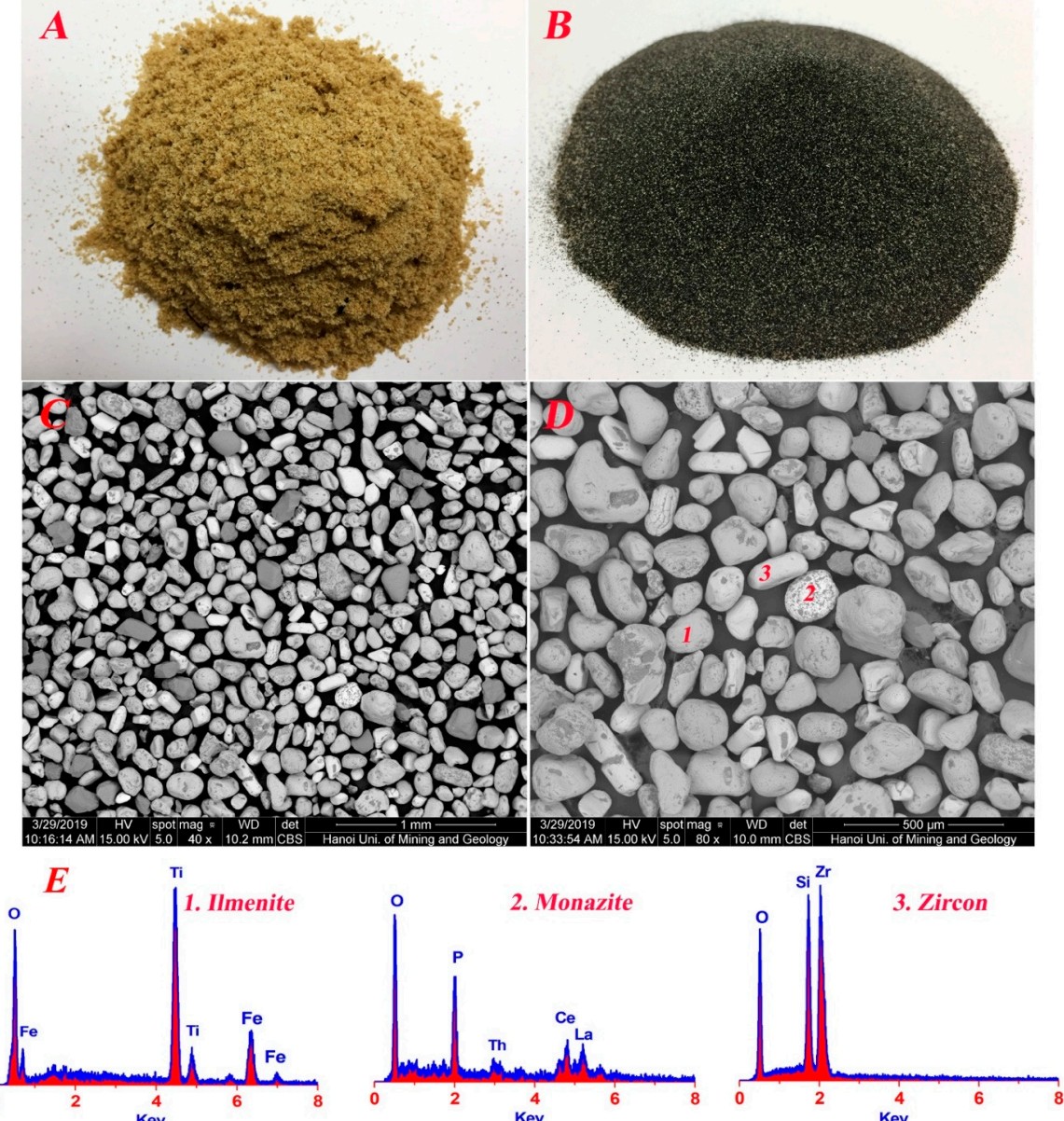

**Figure 5.** Images of the placer sand in the study area. (**A**) Placer sand, (**B**) placer ore, (**C,D**) SEM images of placer ore, (**E**) EDS results of minerals.

In Table 2, the correlation matrix of input and output variables is presented. In the cases of high positive/negative correlations (above approximately 0.75 or below −0.75), they may adversely affect the performance of the models [77]. As a positive sign, all the pairs of attributes in Table 2 are insignificant. Therefore, they have been designated as independent variables to estimate the dependent variable. The feasibility of the AI techniques was assessed by explaining the complicated relationship between predictors and response variable.

**Table 2.** Correlation matrix of heavy minerals.

|  | Rutile | Anatase | Leucoxene | Zircon | Monazite | Ilmenite |
|---|---|---|---|---|---|---|
| Rutile | 1 |  |  |  |  |  |
| Anatase | 0.659115 | 1 |  |  |  |  |
| Leucoxene | 0.516591 | 0.447161 | 1 |  |  |  |
| Zircon | 0.65455 | 0.632258 | 0.326362 | 1 |  |  |
| Monazite | 0.101687 | 0.037384 | 0.084057 | 0.052774 | 1 |  |
| Ilmenite | 0.476926 | 0.650812 | 0.04358 | 0.668171 | −0.00484 | 1 |

## 4. Development of the Model

In order to develop the ilmenite content prediction models, the dataset was separated into two groups according to previous studies [60,78,79]. Specifically, 80% of the original dataset (325 samples) was used to train the models, called the training dataset; 20% remaining of the data (80 samples) was used to validate the model, called the testing dataset. Note that the same sets of training and testing data were used during developing and evaluating of all models. For development of the mentioned model, R software (version 4.5) was used based on their packages.

### 4.1. RF Model

For the RF modeling, the number of trees in the forest (*ntree*) and the randomly selected predictors (*mtry*) were used to adjust the accuracy of the model. The *k*-fold cross-validation method was applied with $k = 10$ to avoid overfitting of the model. According to Nguyen, Bui [80], *ntree* should be selected as 2000 to ensure the forest richness. Subsequently, *mtry* was set from 1 to 50 for finding out the best RF's parameters. Ultimately, the optimal RF model was detected with *mtry* = 2 and *ntree* = 2000, as shown in Figure 6.

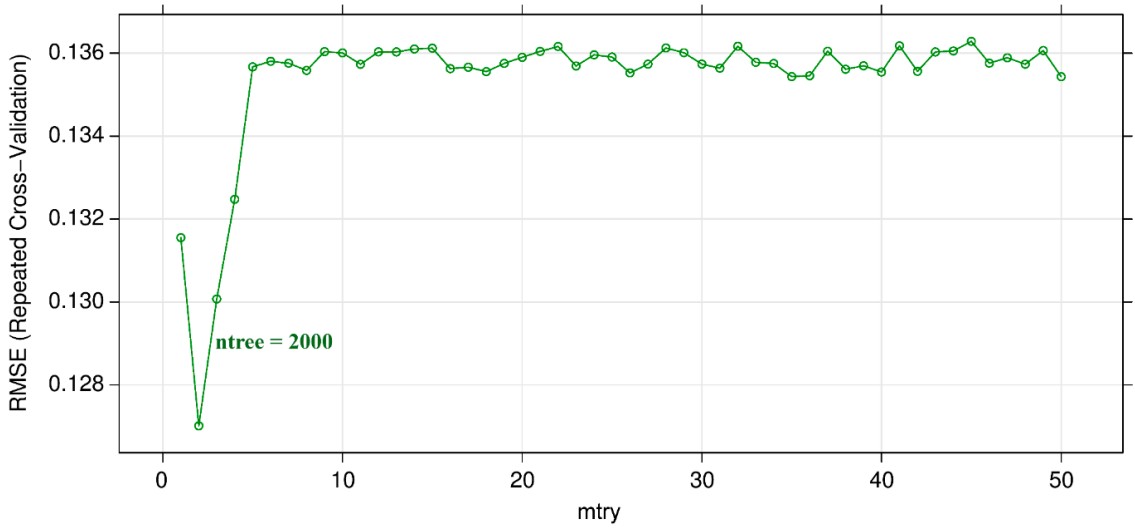

**Figure 6.** Performance of RF model for estimating ilmenite content.

### 4.2. SGB Model

For the SGB model, the boosting iterations ($\alpha$), max tree depth ($\beta$), shrinkage ($\chi$), and minimum terminal node size ($\delta$) were specifically fine-tuned for developing the SGB model. Like the RF model, a 10-fold cross-validation method was also applied to avoid over/underfitting. A trial and error procedure with various parameters of 100 SGB models was performed. The best modelling result was achieved by using $\alpha = 141$, $\beta = 4$, $\chi = 0.1899$, and $\delta = 5$. Figure 7 shows the root-mean-squared error (RMSE) of the SGB model for estimating the content of ilmenite in this study with different parameters.

### 4.3. CART Model

Herein, the optimal CART model was considered and built using only one parameter, which is complexity parameter (*cp*) and a grid search technique with *cp* lies from 0 to 0.1 with interval of 0.002. Numerical experiments showed that *cp* = 0.002 was the best for the CART model in estimating ilmenite content. Figure 8 shows the structure of the CART model for forecasting ilmenite content in this work.

### 4.4. SVM Model

For SVM modelling, various kernel functions can be applied, such as linear, nonlinear, radial basis function (RBF), polynomial, and sigmoid, where RBF is the most-often used one for regression problems [56,61,81,82]. Thus, RBF kernel function was applied for SVM in this study, and $\sigma$ and C (cost) were the significant parameters being fine-turned. There were 100 SVM models using different values of these two parameters performed, and the best result was achieved with $\sigma$ = 0.009 and C = 83.219, as shown in Figure 9.

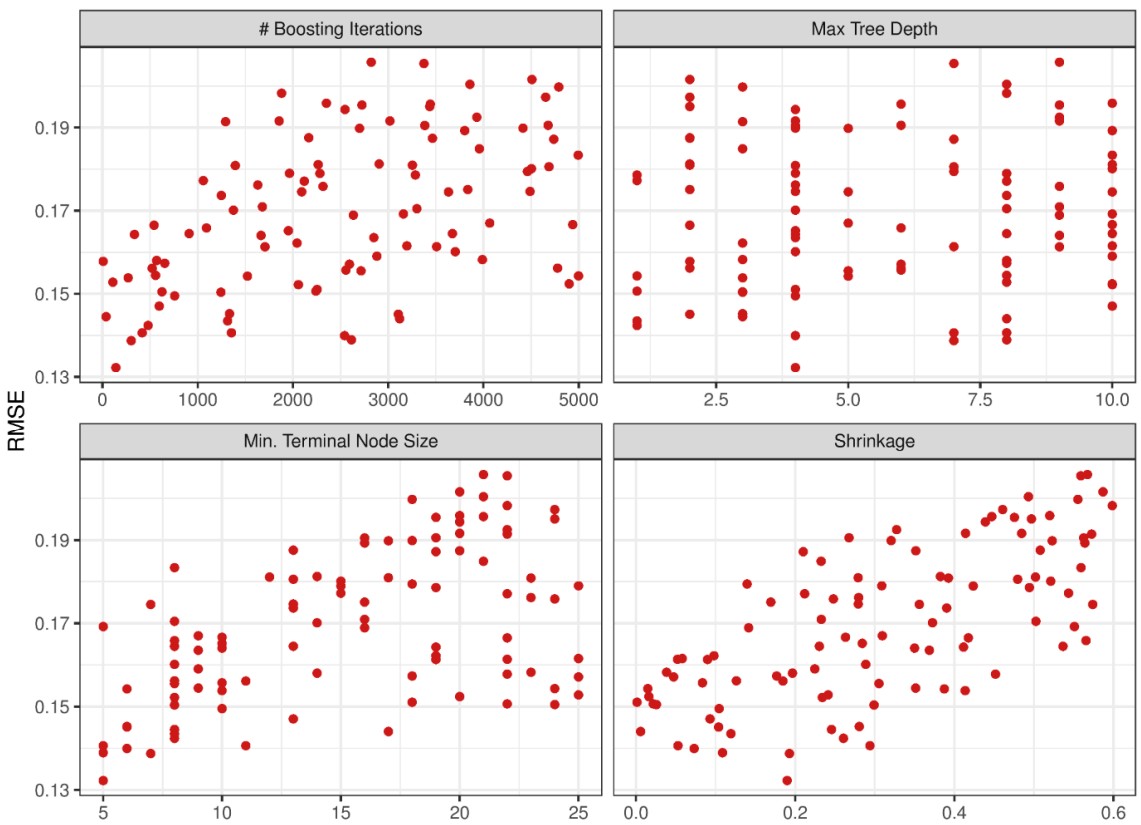

**Figure 7.** Performance of the SGB model in this study.

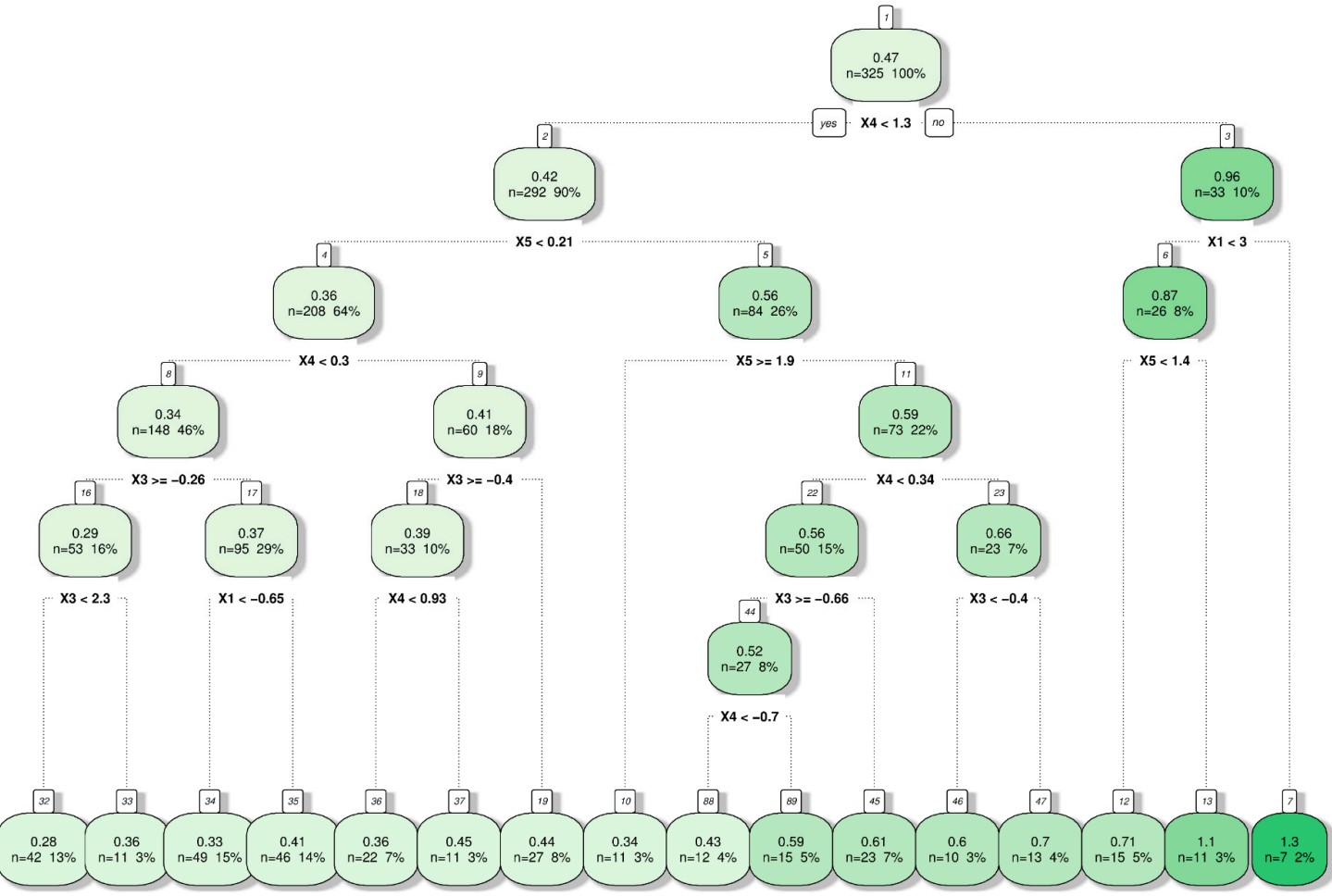

**Figure 8.** The structure of the classification and regression tree (CART) model for estimating ilmenite content.

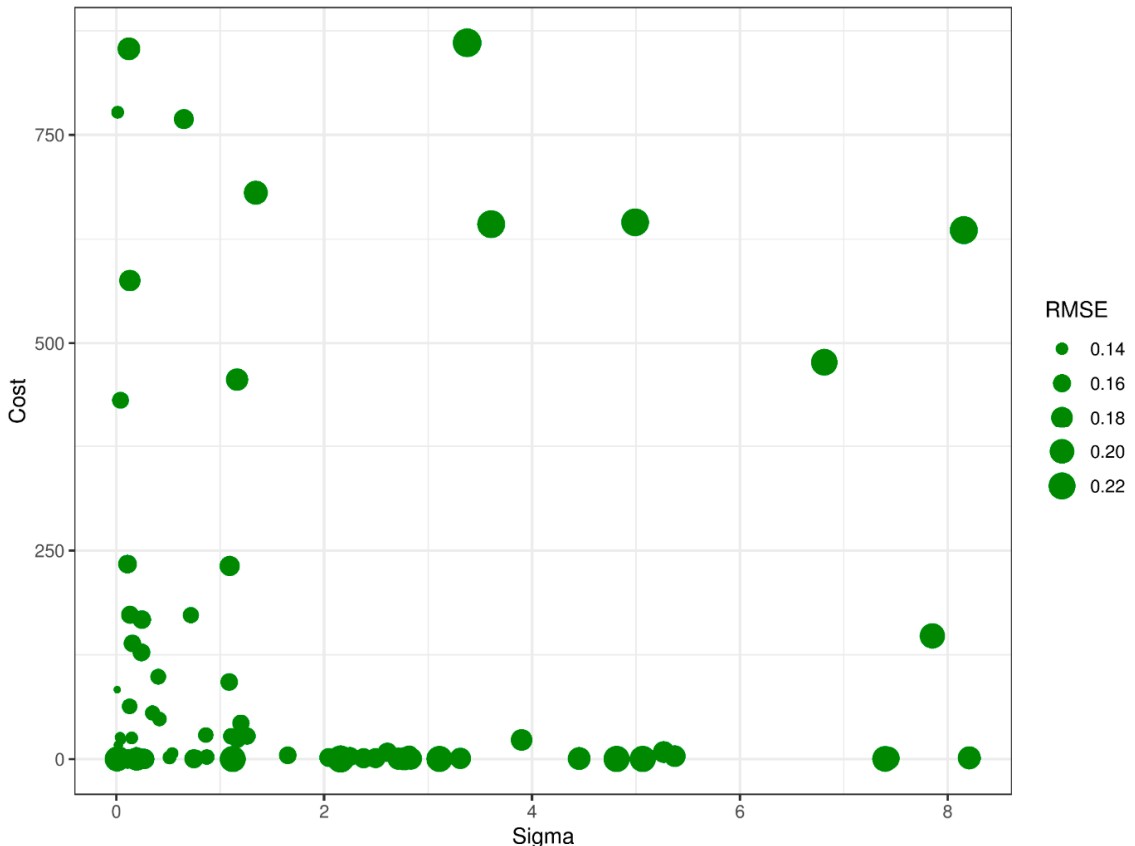

**Figure 9.** Performance of the support vector machine (SVM) model in this study.

### 4.5. Cubist Model

With the cubist model, the rules worked based on the committee ($\varepsilon$) and neighbors/instances ($\phi$). A "trial and error" procedure of $\varepsilon$ and $\phi$ was utilized to find out the best cubist model, as shown in Figure 10. Herein, a grid search technique was applied for $\varepsilon$ and $\phi$ (i.e., $\phi$ = 0 to 9, $\varepsilon$ = 4 to 99). Eventually, the optimal cubist model was determined with $\varepsilon$ = 37 and $\phi$ = 2.

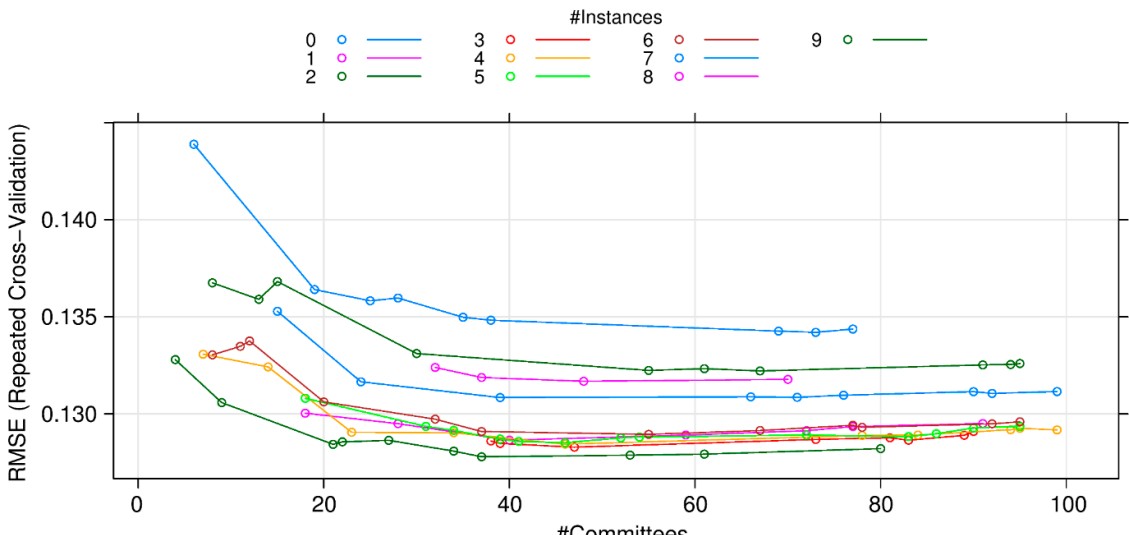

**Figure 10.** Performance of the cubist model in predicting ilmenite content.

### 4.6. The kNN Model

For modeling the ilmenite content by the kNN model, the maximum of the neighbors ($\varphi$), the distance between neighbors ($\gamma$), and kernel function were used to establish the kNN model in this study. Eight kernel functions were applied for the kNN model in this study, including biweight, cosin, epanechnikov, Gaussian, inverse, rectangular, triangular, and triweight. A series of kNN models were tried with various values of $\varphi$ and $\gamma$ to find out the best kNN model. As a result, $\varphi = 71$ and $\gamma = 0.215$ with the inverse kernel function were the best for the kNN model. Figure 11 shows the performance of the kNN model for estimating the content of ilmenite in this study.

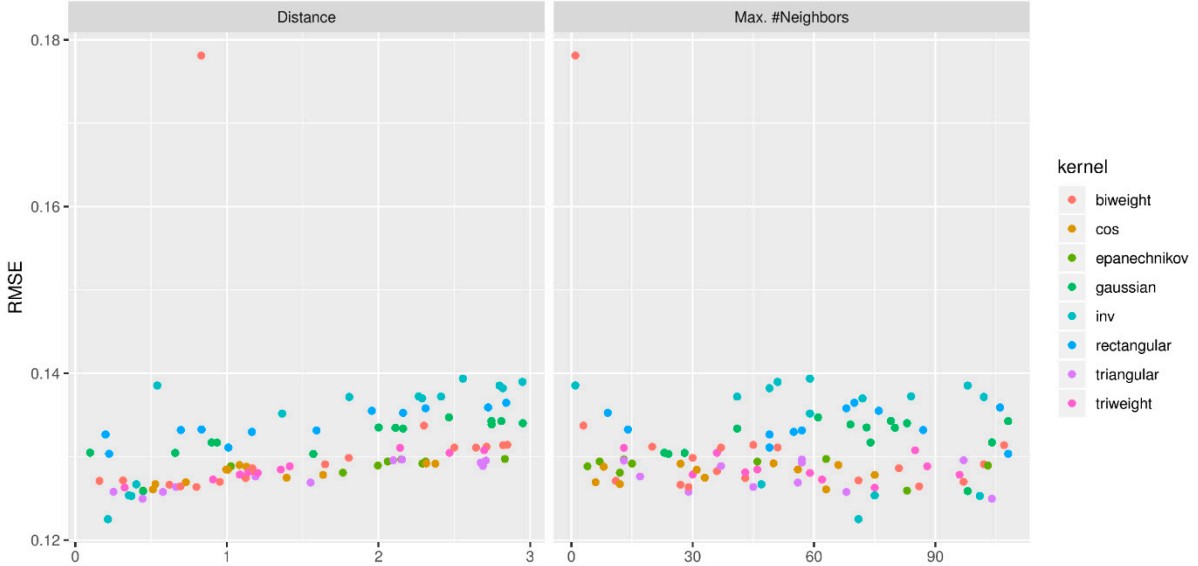

**Figure 11.** Performance of the *k*-nearest neighbors (kNN) model in this study.

### 4.7. ANN Model

For ANN modelling, the hidden layers and neurons per layer need to be chosen carefully. Too many hidden layers or neurons can lead to overfitting [83]. In addition, they can affect the processing time of the model. Therefore, ANNs with two or three hidden layers were recommended by Nguyen et al. [61] for the simple regression problems. In this study, an ANN model with a structure with two hidden layers was selected to estimate the ilmenite content. Its structure includes 5, 16, 10, and 1 neurons, for the input, the first and second hidden, and output layers, respectively (Figure 12). Unlike the previous models, the min-max scale method was applied to normalize the dataset in the range of (−1 to 1); 50 repetitions were performed to determine the initial weights and biases of the ANN model. Subsequently, the optimal weights and biases of the ANN model were calculated, as shown in Figure 12 through the black and grey lines.

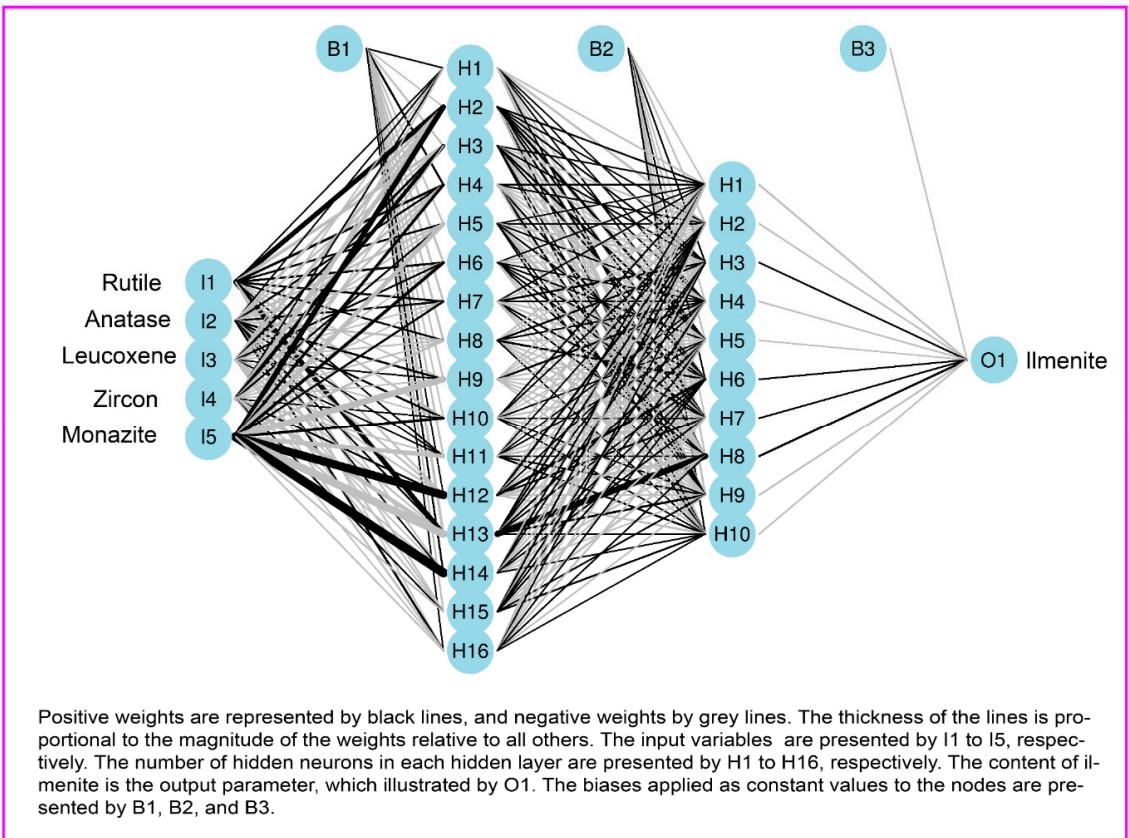

Positive weights are represented by black lines, and negative weights by grey lines. The thickness of the lines is proportional to the magnitude of the weights relative to all others. The input variables are presented by I1 to I5, respectively. The number of hidden neurons in each hidden layer are presented by H1 to H16, respectively. The content of ilmenite is the output parameter, which illustrated by O1. The biases applied as constant values to the nodes are presented by B1, B2, and B3.

**Figure 12.** Artificial neural network (ANN) model for estimating the content of ilmenite.

## 5. Performance Indicators for Evaluating the Soft Computing Techniques

In the present study, root-mean-squared error (RMSE), coefficient of correlation ($R^2$), ranking method, and residuals analysis technique were used to rate the models' quality. The performance indicators are computed as follow:

$$\text{RMSE} = \sqrt{\frac{1}{n}\sum_{i=1}^{n} \left(y_{i.ilmenite} - \hat{y}_{i.ilmenite}\right)^2} \tag{4}$$

$$R^2 = 1 - \frac{\sum_{i} \left(y_{i.ilmenite} - \hat{y}_{i.ilmenite}\right)^2}{\sum_{i} \left(y_{i.ilmenite} - \overline{y}\right)^2} \tag{5}$$

where $n$ is a total number of experimental datasets; $y_{i.ilmenite}$, $\hat{y}_{i.ilmenite}$ and $\overline{y}$ are measured, predicted, and mean of $y_i$ values, respectively.

## 6. Results and Discussions

After developing the models, their performance was compared and evaluated using RMSE and $R^2$, ranking, and residual plots being analyzed. Based on the training dataset, the ilmenite content predictive models (i.e., SVM, RF, SGB, CART, kNN, cubist, and ANN) were developed as described above. Table 3 calculates the indicators of performance of the soft computing techniques for estimating the content of ilmenite in the training process.

**Table 3.** Performance of the soft computing techniques for estimating ilmenite content (training dataset).

| Model | RMSE | $R^2$ | Rank for RMSE | Rank for $R^2$ | Total Ranking Score | Sort |
|---|---|---|---|---|---|---|
| SVM | 0.134 | 0.692 | 2 | 2 | 4 | 6 |
| CART | 0.147 | 0.675 | 1 | 1 | 2 | 7 |
| kNN | 0.122 | 0.747 | 6 | 6 | 12 | 2 |
| RF | 0.127 | 0.737 | 5 | 5 | 10 | 3 |
| SGB | 0.132 | 0.716 | 3 | 3 | 6 | 5 |
| Cubist | 0.128 | 0.720 | 4 | 4 | 8 | 4 |
| ANN | 0.091 | 0.860 | 7 | 7 | 14 | 1 |

Performance of the developed soft computing models in Table 3 showed that all the AI models were able to generate reasonably accurate ilmenite content estimation. Most models provided good performance with $R^2$ over 0.7 where some achieved $R^2$ over 0.8. Additionally, based on the ranking index, the ANN is the best model on the training dataset. To verify the performances of the stated soft computing techniques, 80 experimental datasets on the set of testing data was used as described above. Model validation is considered as the most critical step in the model building sequence. Table 4 computes the performance on the models using the testing dataset based on performance indicators.

**Table 4.** Performance of the soft computing techniques for estimating ilmenite content (testing dataset).

| Model | RMSE | $R^2$ | Rank for RMSE | Rank for $R^2$ | Total Ranking Score | Sort |
|---|---|---|---|---|---|---|
| SVM | 0.092 | 0.780 | 1 | 3 | 4 | 5 |
| CART | 0.082 | 0.817 | 4 | 4 | 8 | 4 |
| kNN | 0.092 | 0.766 | 1 | 1 | 2 | 7 |
| RF | 0.080 | 0.824 | 6 | 6 | 12 | 2 |
| SGB | 0.081 | 0.818 | 5 | 5 | 10 | 3 |
| Cubist | 0.078 | 0.830 | 7 | 7 | 14 | 1 |
| ANN | 0.092 | 0.774 | 1 | 2 | 3 | 6 |

From Table 4, there are some differences from Table 3. Whereas the ANN is the best soft computing model on the training dataset, the cubist model with the highest performances, as well as the ranking, became the best model on the testing dataset. At the other end, the kNN model with the lowest performances and ranking on validation set also became the worst model on the testing dataset. Notably, although the ANN model on the training data set was the top scorer, its performance decreased significantly on the test set. It showed the unstable nature of ANN model in estimating the ilmenite content in this study. The RF and SGB models still retained the same stable performance as on the training dataset. Figure 13 shows the accuracy of the developed models on the testing dataset through $R^2$ values.

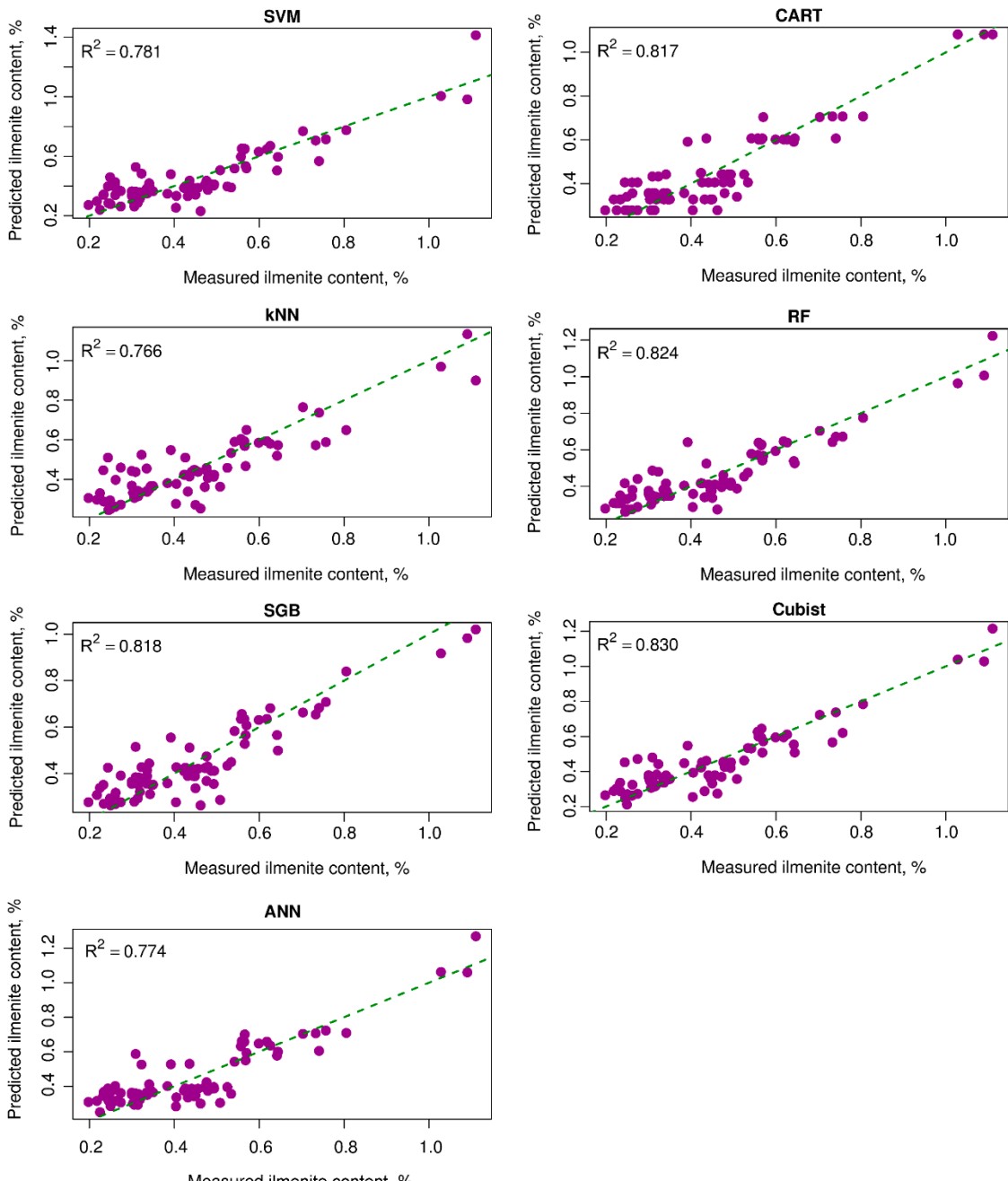

**Figure 13.** Predicted/measured parameters of the soft computing techniques used herein.

Although the R² values of the models were high, they did not guarantee that the models fit the data well. Therefore, the residuals analysis of the models was conducted in order to check the assumptions of independence, normality, and homoscedasticity. The histogram (Figure 14) and the normal probability plots (Figure 15) were used to check whether or not it is justifiable to suppose that the random errors inherent in the developed soft computing models were extracted from a normal distribution.

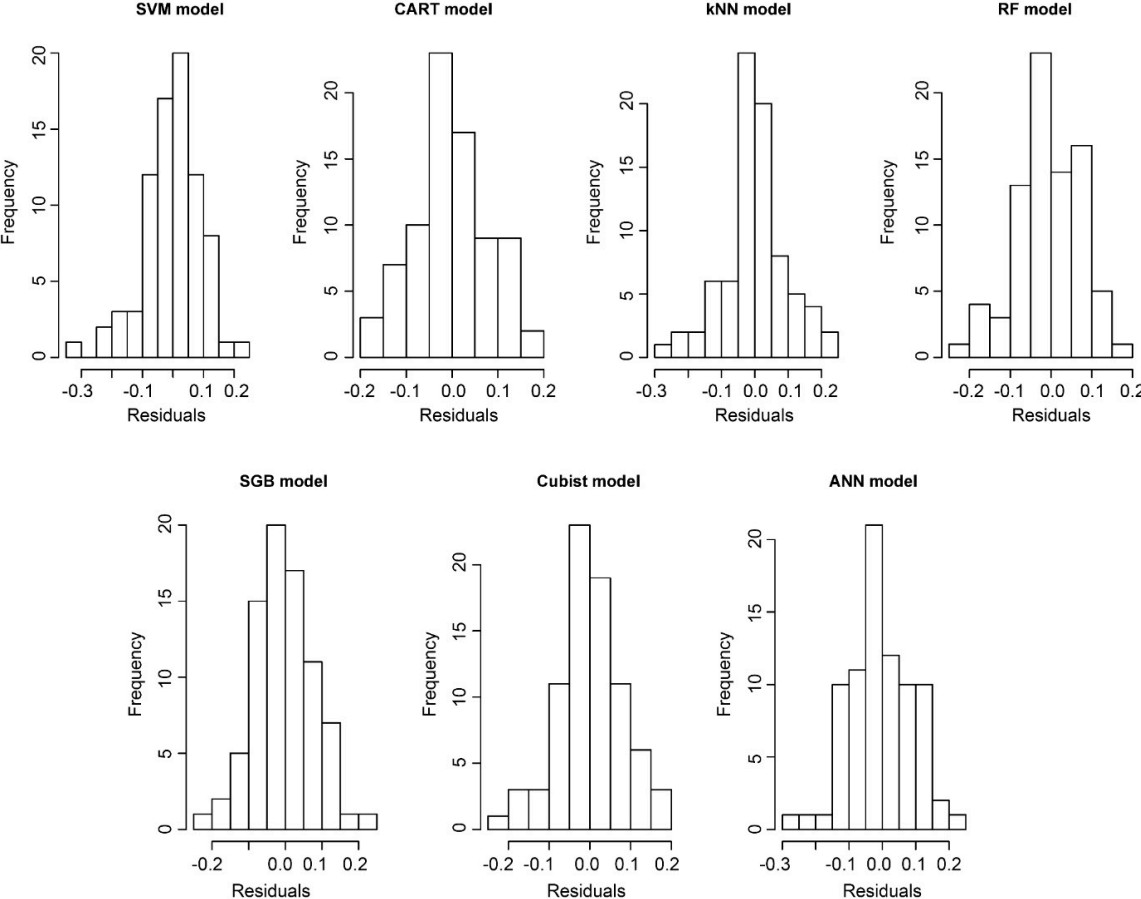

**Figure 14.** Histogram plots of the residuals of the models.

Based on histogram plots of the residuals of the models (Figure 14), we can see that most of the residuals of the models are Gaussian or normal distribution. It suggested that the models fit the data well. Notably, the residuals of the SGB and cubist models seem to be better fitted to the normal distribution. Additionally, the frequency of the SGB model appears to be more stable than the cubist model. The residuals of the SGB model are smaller and approximate random errors. These points showed that the SGB model seems to be the most suitable model for the data used in this study.

Furthermore, the quantile-quantile plots (Q-Q plots) were used (Figure 15) in order to confirm the normality of the data for the developed models. Figure 15 shows the normality of the residuals of the models visually. If the SGB model and cubist model were good candidates for the data of this study in Figure 14, then, in Figure 15, the SGB model shows that it was more suitable for the data than the cubist model. Based on the results of Tables 3 and 4 and the residual analysis of the models (Figures 14 and 15), the SGB model should be selected as the best soft computing technique for estimating the ilmenite content in this study.

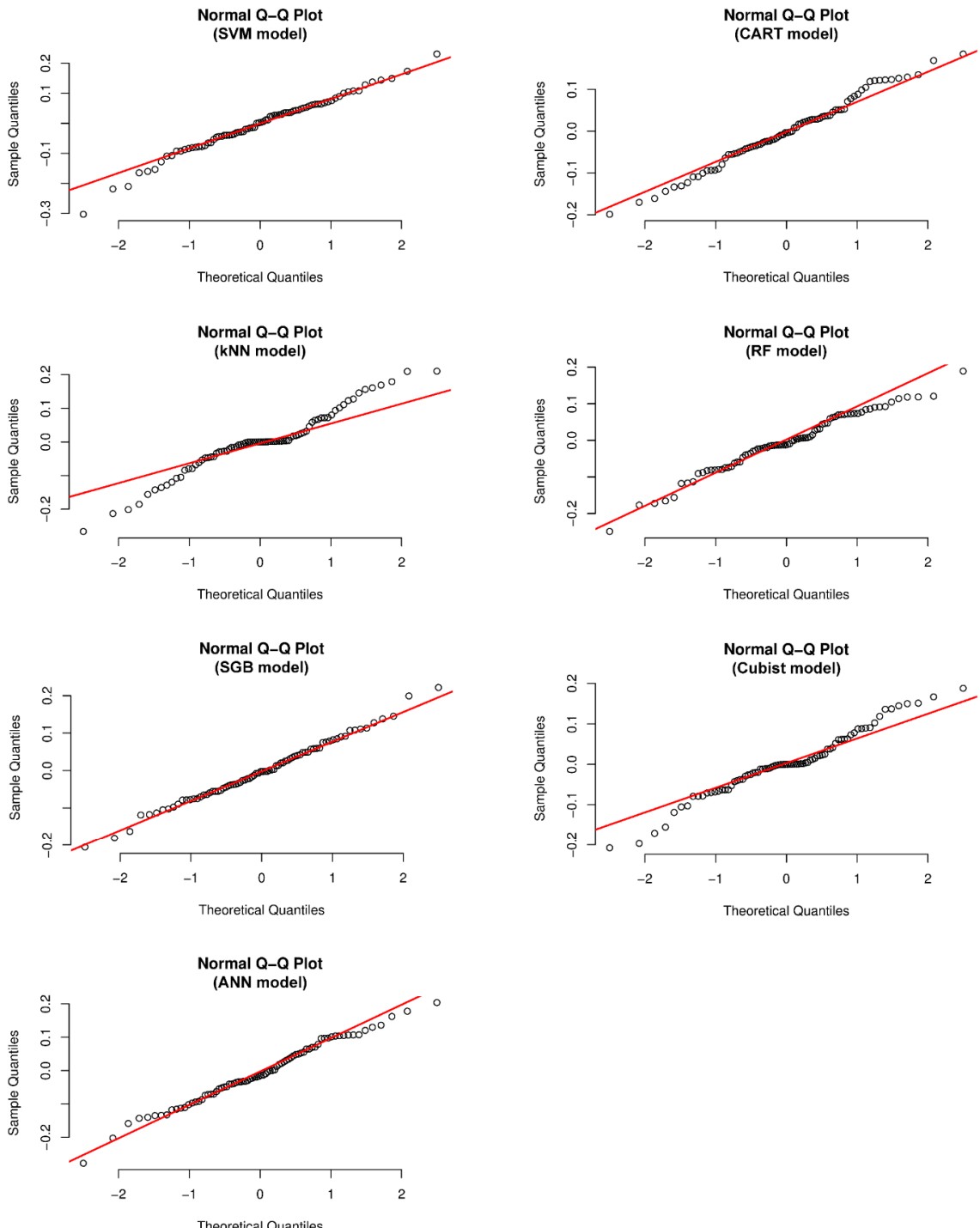

**Figure 15.** Normal probability plots of the residuals of the soft computing techniques herein.

## 7. Conclusions

Ilmenite is a fairly common and industrially valuable titanium-containing mineral in coastal placer mines, including in Vietnam. In this study, 405 samples from the Suoi Nhum mine, Vietnam, were collected and processed to separate different heavy minerals. The weight percents of each heavy mineral (rutile, anatase, leucoxene, zircon, and monazite) were used as input data for predicting ilmenite content by AI techniques. The obtained results of this work indicated that ilmenite content and the remaining other heavy minerals are highly correlated.

As a conclusion, AI is a robust technique that can be applied in practical engineering to determine the content of ilmenite in titanium placer/beach placer sand with a significant reliability. This study demonstrated that the SGB model is the best model for estimating ilmenite content. Additionally, the cubist model can also be used in practical engineering. The remaining models (kNN, RF, SVM, CART, and ANN) were also considered in other conditions or areas. It helps to investigate and define potential heavy mineral areas more appropriately. On the other hand, the results of this research are the basis for selecting the mineral mining areas in the placer mines more reasonably.

**Author Contributions:** Data collection and experimental works: H.N., H.-B.B., and Q.-T.L. Writing, discussion, analysis, and revision: H.N., H.-B.B., Q.-T.L., Y.L., X.-N.B., T.N.-T., J.D., and X.S. All authors have read and agreed to the published version of the manuscript.

**Funding:** This work received no external funding.

**Acknowledgments:** The authors would like to thank Hanoi University of Mining and Geology (HUMG), Hanoi, Vietnam; the Center for Excellence in Analysis and Experiment and the Center for Mining, Electro-Mechanical research of HUMG; Duy Tan University, Da Nang, Vietnam, and Ton Duc Thang University, Ho Chi Minh City, Vietnam.

**Conflicts of Interest:** The authors declare no conflicts of interest.

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
