# Peer review of "A Comparative Study of Different Machine Learning Algorithms in Predicting the Content of Ilmenite in Titanium Placer"

_applsci, doi:10.3390/app10020635_

Round 1
Reviewer 1 Report
This is an interesting contribution for using AI in the field of Mining and Mineralogy.
Before of accepting the ms, I consider that authors must clarify:
If other instrumental techniques apart from SEM were used to determine the mineralogical variables. How were used SEM images and EDX analyses as input for the IA processes.Author Response
Please see the attachment.

Reviewer 2 Report
I think the paper presents a novel and interesting approach to aid in mineral resource evaluation of titanium placers and possible be applied to other placer deposit types and different economic mineral resource systems. Critical to evaluating the AI methods and results is understanding what parameters and variables were used in each method. The input isn't clear.
How was the ilmenite content derived? Was there actually any ilmenite, titanomagnetite, or titanium component to any magnetite in the placer? How was the ilmenite component calculated for rutile, anatase, leucoxene, zircon and monazite? Why abbreviate X1, X2, X3, X4, X5 and Y, when the name itself is just as easy. Its not clear what part of these data collected from the samples was integrated into each method and how? The AI methods and statistics are clear but how the geology and mineral chemistry is applied needs to be explained in a context that the reader can duplicate a method with your data and arrive at the same result, or apply their own data for comparison.These issues need to be addressed before the results and application of methods can be critically evaluated.

Reviewer 3 Report
Review report of A Comparative Study of Different Machine Learning Algorithms in Predicting the Content of Ilmenite in Titanium Placer by Nguyen et al.,
In general there is not a clear explanation of the geological context of the study for reader. I find it hard to relate this computing techniques to estimate a certain amount of Ilmenite in a specific area. I suggest you to give more information about your samples, mineralogy of samples and how you detect the ilmenite content. Of course you use cutting edge techniques but for me it is too focused on data handling itself, but to gain attention of mineral explorers, you can highlight more of the actual geological samples.
I found it hard to track to relationship between your actual "hard" sample properties with your data handling procedures. Are your samples are really homogenous and applicable that kind of work? and what exactly did you find and how can it be extrapolated in the other similar TiO2 rich placer deposits of the world?
In the attached file, you can found specific comments embedded in the lines.
Best regards

Round 2
Reviewer 1 Report
The authors have considered and explained my previous questions.
Author Response
Thank you very much!
Reviewer 2 Report
The AI methods and statistics are clear but how the geology and mineral components are applied needs to be explained in a context that the reader can duplicate a method with your data and arrive at the same result, or apply their own data for comparison.
The authors need to show how the mineral components were used as inputs for each model. Rather than showing a generalized or pseudo-code model, show how the heavy mineral separate values were integrated.
These issues need to be addressed before the results and application of methods can be critically evaluated.

Round 3
Reviewer 2 Report
I would still suggest more information explaining how the placer weight percents are included in the analyses. Other than that, where a reader could reproduce and duplicate these methods with other resource types, the paper appears ready.